# GENESIS: GENERATIVE SCENE INFERENCE AND SAMPLING WITH OBJECT-CENTRIC LATENT REPRESENTATIONS

**Martin Engelcke**$^{*\nabla}$**, Adam R. Kosiorek**$^{\nabla\triangle}$**, Oiwi Parker Jones**$^{\nabla}$ **& Ingmar Posner**$^{\nabla}$
$^{\nabla}$ Applied AI Lab, University of Oxford; $^{\triangle}$ Dept. of Statistics, University of Oxford

## ABSTRACT

Generative latent-variable models are emerging as promising tools in robotics and reinforcement learning. Yet, even though tasks in these domains typically involve distinct objects, most state-of-the-art generative models do not explicitly capture the compositional nature of visual scenes. Two recent exceptions, MONet and IODINE, decompose scenes into objects in an unsupervised fashion. Their underlying generative processes, however, do not account for component interactions. Hence, neither of them allows for principled sampling of novel scenes. Here we present GENESIS, the first object-centric generative model of rendered 3D scenes capable of both decomposing *and* generating scenes by capturing relationships between scene components. GENESIS parameterises a spatial GMM over images which is decoded from a set of object-centric latent variables that are either inferred sequentially in an amortised fashion or sampled from an autoregressive prior. We train GENESIS on several publicly available datasets and evaluate its performance on scene generation, decomposition, and semi-supervised learning.

## 1 INTRODUCTION

Task execution in robotics and reinforcement learning (RL) requires accurate perception of and reasoning about discrete elements in an environment. While supervised methods can be used to identify pertinent objects, it is intractable to collect labels for every scenario and task. Discovering structure in data—such as objects—and learning to represent data in a compact fashion without supervision are long-standing problems in machine learning (Comon, 1992; Tishby et al., 2000), often formulated as *generative latent-variable modelling* (e.g. Kingma & Welling, 2014; Rezende et al., 2014). Such methods have been leveraged to increase sample efficiency in RL (Gregor et al., 2019) and other supervised tasks (van Steenkiste et al., 2019). They also offer the ability to *imagine* environments for training (Ha & Schmidhuber, 2018). Given the compositional nature of visual scenes, separating latent representations into object-centric ones can facilitate fast and robust learning (Watters et al., 2019a), while also being amenable to relational reasoning (Santoro et al., 2017). Interestingly, however, state-of-the-art methods for generating realistic images do not account for this discrete structure (Brock et al., 2018; Parmar et al., 2018).

As in the approach proposed in this work, human visual perception is not passive. Rather it involves a creative interplay between external stimulation and an active, internal generative model of the world (Rao & Ballard, 1999; Friston, 2005). That this is necessary can be seen from the physiology of the eye, where the small portion of the visual field that can produce sharp images (*fovea centralis*) motivates the need for rapid eye movements (*saccades*) to build up a crisp and holistic percept of a scene (Wandell, 1995). In other words, what we perceive is largely a mental simulation of the external world. Meanwhile, work in computational neuroscience tells us that visual features (see, e.g., Hubel & Wiesel, 1968) can be inferred from the statistics of static images using unsupervised learning (Olshausen & Field, 1996). Experimental investigations further show that specific brain areas (e.g. LO) appear specialised for objects, for example responding more strongly to common objects than to scenes or textures, while responding only weakly to movement (cf. MT) (e.g., Grill-Spector & Malach, 2004).

---

$^{*}$Corresponding author: `martin@robots.ox.ac.uk`

In this work, we are interested in probabilistic generative models that can explain visual scenes compositionally via several latent variables. This corresponds to fitting a probability distribution $p_\theta(\mathbf{x})$ with parameters $\theta$ to the data. The compositional structure is captured by $K$ latent variables so that $p_\theta(\mathbf{x}) = \int p_\theta(\mathbf{x} \mid \mathbf{z}_{1:K}) p_\theta(\mathbf{z}_{1:K}) \, \mathrm{d}\mathbf{z}_{1:K}$. Models from this family can be optimised using the variational auto-encoder (VAE) framework (Kingma & Welling, 2014; Rezende et al., 2014), by maximising a variational lower bound on the model evidence (Jordan et al., 1999). Burgess et al. (2019) and Greff et al. (2019) recently proposed two such models, MONet and IODINE, to decompose visual scenes into meaningful objects. Both works leverage an *analysis-by-synthesis* approach through the machinery of VAEs (Kingma & Welling, 2014; Rezende et al., 2014) to train these models without labelled supervision, e.g. in the form of ground truth segmentation masks. However, the models have a factorised prior that treats scene components as independent. Thus, neither provides an object-centric generation mechanism that accounts for relationships between constituent parts of a scene, e.g. two physical objects cannot occupy the same location, prohibiting the component-wise generation of novel scenes and restricting the utility of these approaches. Moreover, MONet embeds a convolutional neural network (CNN) inside of an recurrent neural network (RNN) that is unrolled for each scene component, which does not scale well to more complex scenes. Similarly, IODINE utilises a CNN within an expensive, gradient-based iterative refinement mechanism.

Therefore, we introduce GENErative Scene Inference and Sampling (GENESIS) which is, to the best of our knowledge, the first object-centric generative model of rendered 3D scenes capable of both decomposing and generating scenes[1]. Compared to previous work, this renders GENESIS significantly more suitable for a wide range of applications in robotics and reinforcement learning. GENESIS achieves this by modelling relationships between scene components with an expressive, autoregressive prior that is learned alongside a sequential, amortised inference network. Importantly, sequential inference is performed in low-dimensional latent space, allowing all convolutional encoders and decoders to be run in parallel to fully exploit modern graphics processing hardware.

We conduct experiments on three canonical and publicly available datasets: *coloured Multi-dSprites* (Burgess et al., 2019), the *GQN* dataset (Eslami et al., 2018), and *ShapeStacks* (Groth et al., 2018). The latter two are simulated 3D environments which serve as testing grounds for navigation and object manipulation tasks, respectively. We show both qualitatively and quantitatively that in contrast to prior art, GENESIS is able to generate coherent scenes while also performing well on scene decomposition. Furthermore, we use the scene annotations available for ShapeStacks to show the benefit of utilising general purpose, object-centric latent representations from GENESIS for tasks such as predicting whether a block tower is stable or not.

Code and models are available at `https://github.com/applied-ai-lab/genesis`.

## 2 RELATED WORK

**Structured Models** Several methods leverage structured latent variables to discover objects in images without direct supervision. CST-VAE (Huang & Murphy, 2015), AIR (Eslami et al., 2016), SQAIR (Kosiorek et al., 2018), and SPAIR (Crawford & Pineau, 2019) use spatial attention to partition scenes into objects. TAGGER (Greff et al., 2016), NEM (Greff et al., 2017), and R-NEM (van Steenkiste et al., 2018a) perform unsupervised segmentation by modelling images as spatial mixture models. SCAE (Kosiorek et al., 2019) discovers geometric relationships between objects and their parts by using an affine-aware decoder. Yet, these approaches have not been shown to work on more complex images, for example visual scenes with 3D spatial structure, occlusion, perspective distortion, and multiple foreground and background components as considered in this work. Moreover, none of them demonstrate the ability to generate novel scenes with relational structure.

While Xu et al. (2018) present an extension of Eslami et al. (2016) to generate images, their method only works on binary images with a uniform black background and assumes that object bounding boxes do not overlap. In contrast, we train GENESIS on rendered 3D scenes from Eslami et al. (2018) and Groth et al. (2018) which feature complex backgrounds and considerable occlusion to perform both decomposition *and* generation. Lastly, Xu et al. (2019) use ground truth pixel-wise flow fields as a cue for segmenting objects or object parts. Similarly, GENESIS could be adapted to also leverage temporal information which is a promising avenue for future research.

---

[1] We use the terms "object" and "scene component" synonymously in this work.

MONet & IODINE   While this work is most directly related to MONet (Burgess et al., 2019) and IODINE (Greff et al., 2019), it sets itself apart by introducing a generative model that captures relations between scene components with an autoregressive prior, enabling the unconditional generation of coherent, novel scenes. Moreover, MONet relies on a deterministic attention mechanism rather than utilising a proper probabilistic inference procedure. This implies that the training objective is not a valid lower bound on the marginal likelihood and that the model cannot perform density estimation without modification. Furthermore, this attention mechanism embeds a CNN in a RNN, posing an issue in terms of scalability. These two considerations do not apply to IODINE, but IODINE employs a gradient-based, iterative refinement mechanism which expensive both in terms of computation and memory, limiting its practicality and utility. Architecturally, GENESIS is more similar to MONet and does not require expensive iterative refinement as IODINE. Unlike MONet, though, the convolutional encoders and decoders in GENESIS can be run in parallel, rendering the model computationally more scalable to inputs with a larger number of scene components.

**Adversarial Methods**   A few recent works have proposed to use an adversary for scene segmentation and generation. Chen et al. (2019) and Bielski & Favaro (2019) segment a single foreground object per image and Arandjelović & Zisserman (2019) segment several synthetic objects superimposed on natural images. Azadi et al. (2019) combine two objects or an object and a background scene in a sensible fashion and van Steenkiste et al. (2018b) can generate scenes with a potentially arbitrary number of components. In comparison, GENESIS performs both inference and generation, does not exhibit the instabilities of adversarial training, and offers a probabilistic formulation which captures uncertainty, e.g. during scene decomposition. Furthermore, the complexity of GENESIS increases with $\mathcal{O}(K)$, where $K$ is the number of components, as opposed to the $\mathcal{O}(K^2)$ complexity of the *relational stage* in van Steenkiste et al. (2018b).

**Inverse Graphics**   A range of works formulate scene understanding as an inverse graphics problem. These well-engineered methods, however, rely on scene annotations for training and lack probabilistic formulations. For example, Wu et al. (2017b) leverage a graphics renderer to decode a structured scene description which is inferred by a neural network. Romaszko et al. (2017) pursue a similar approach but instead make use of a differentiable graphics render. Wu et al. (2017a) further employ different physics engines to predict the movement of billiard balls and block towers.

## 3   GENESIS: GENERATIVE SCENE INFERENCE AND SAMPLING

In this section, we first describe the generative model of GENESIS and a simplified variant called GENESIS-S. This is followed by the associated inference procedures and two possible learning objectives. GENESIS is illustrated in Figure 1 and Figure 2 shows the graphical model in comparison to alternative methods. An illustration of GENESIS-S is included Appendix B.1, Figure 5.

**Generative model**   Let $\mathbf{x} \in \mathbb{R}^{H \times W \times C}$ be an image. We formulate the problem of image generation as a spatial Gaussian mixture model (GMM). That is, every Gaussian component $k = 1, \ldots, K$ represents an image-sized scene component $\mathbf{x}_k \in \mathbb{R}^{H \times W \times C}$. $K \in \mathbb{N}_+$ is the maximum number of scene components. The corresponding *mixing probabilities* $\pi_k \in [0, 1]^{H \times W}$ indicate whether the component is present at a location in the image. The mixing probabilities are normalised across scene components, i.e. $\forall_{i,j} \sum_k \pi_{i,j,k} = 1$, and can be regarded as spatial *attention masks*. Since there are strong spatial dependencies between components, we formulate an autoregressive prior distribution over mask variables $\mathbf{z}_k^m \in \mathbb{R}^{D_m}$ which encode the mixing probabilities $\pi_k$, as

$$p_\theta(\mathbf{z}_{1:K}^m) = \prod_{k=1}^{K} p_\theta(\mathbf{z}_k^m \mid \mathbf{z}_{1:k-1}^m) = \prod_{k=1}^{K} p_\theta(\mathbf{z}_k^m \mid \mathbf{u}_k)|_{\mathbf{u}_k = \mathrm{R}_\theta(\mathbf{z}_{k-1}^m, \mathbf{u}_{k-1})}. \tag{1}$$

The dependence on previous latents $\mathbf{z}_{1:k-1}^m$ is implemented via an RNN $\mathrm{R}_\theta$ with hidden state $\mathbf{u}_k$.

Next, we assume that the scene components $\mathbf{x}_k$ are conditionally independent given their spatial allocation in the scene. The corresponding conditional distribution over component variables $\mathbf{z}_k^c \in \mathbb{R}^{D_c}$ which encode the scene components $\mathbf{x}_k$ factorises as follows,

$$p_\theta(\mathbf{z}_{1:K}^c \mid \mathbf{z}_{1:K}^m) = \prod_{k=1}^{K} p_\theta(\mathbf{z}_k^c \mid \mathbf{z}_k^m). \tag{2}$$

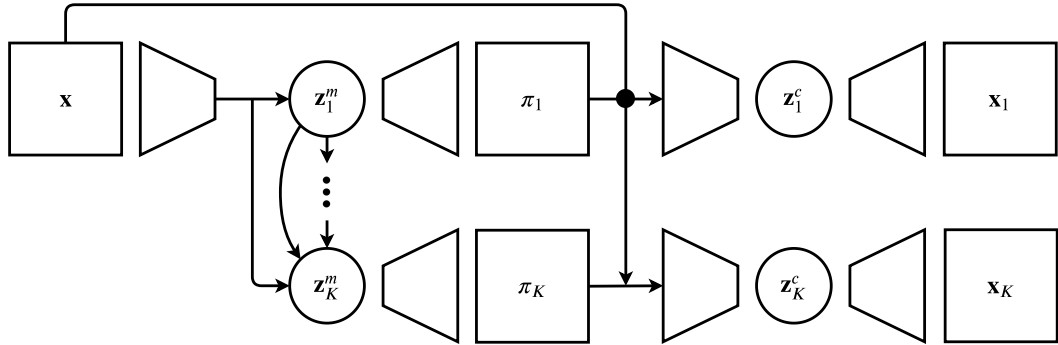

Figure 1: GENESIS illustration. Given an image $\mathbf{x}$, an encoder and an RNN compute the mask latents $\mathbf{z}_k^m$. These are decoded to obtain the mixing probabilities $\pi_k$. The image and individual masks are concatenated to infer the component latents $\mathbf{z}_k^c$ from which the scene components $\mathbf{x}_k$ are decoded.

Now, the image likelihood is given by a mixture model,

$$p(\mathbf{x} \mid \mathbf{z}_{1:K}^m, \mathbf{z}_{1:K}^c) = \sum_{k=1}^{K} \pi_k \, p_\theta(\mathbf{x}_k \mid \mathbf{z}_k^c) \,, \tag{3}$$

where the mixing probabilities $\pi_k = \pi_\theta(\mathbf{z}_{1:k}^m)$ are created via a stick-breaking process (SBP) adapted from Burgess et al. (2019) as follows, slightly overloading the $\pi$ notation,

$$\pi_1 = \pi_\theta(\mathbf{z}_1^m)\,, \qquad \pi_k = \left(1 - \sum_{j=1}^{k-1} \pi_j\right) \pi_\theta(\mathbf{z}_k^m)\,, \qquad \pi_K = \left(1 - \sum_{j=1}^{K-1} \pi_j\right) . \tag{4}$$

Note that this step is not necessary for our model and instead one could use a $\mathrm{softmax}$ to normalise masks as in Greff et al. (2019).

Finally, omitting subscripts, the full generative model can be written as

$$p_\theta(\mathbf{x}) = \iint p_\theta(\mathbf{x} \mid \mathbf{z}^c, \mathbf{z}^m) p_\theta(\mathbf{z}^c \mid \mathbf{z}^m) p_\theta(\mathbf{z}^m) \, \mathrm{d}\mathbf{z}^m \, \mathrm{d}\mathbf{z}^c \,, \tag{5}$$

where we assume that all conditional distributions are Gaussian. The Gaussian components of the image likelihood have a fixed scalar standard deviation $\sigma_x^2$. We refer to this model as GENESIS. To investigate whether separate latents for masks and component appearances are necessary for decomposition, we consider a simplified model, GENESIS-S, with a single latent variable per component,

$$p_\theta(\mathbf{z}_{1:K}) = \prod_{k=1}^{K} p_\theta(\mathbf{z}_k \mid \mathbf{z}_{1:k-1}). \tag{6}$$

In this case, $\mathbf{z}_k$ takes the role of $\mathbf{z}_k^c$ in Equation (3) and of $\mathbf{z}_k^m$ in Equation (4), while Equation (2) is no longer necessary.

**Approximate posterior**  We amortise inference by using an approximate posterior distribution with parameters $\phi$ and a structure similar to the generative model. The full approximate posterior reads as follows,

$$q_\phi(\mathbf{z}_{1:K}^c, \mathbf{z}_{1:K}^m \mid \mathbf{x}) = q_\phi(\mathbf{z}_{1:K}^m \mid \mathbf{x}) \, q_\phi(\mathbf{z}_{1:K}^c \mid \mathbf{x}, \mathbf{z}_{1:K}^m)\,, \quad \text{where}$$

$$q_\phi(\mathbf{z}_{1:K}^m \mid \mathbf{x}) = \prod_{k=1}^{K} q_\phi\big(\mathbf{z}_k^m \mid \mathbf{x}, \mathbf{z}_{1:k-1}^m\big)\,, \quad \text{and} \quad q_\phi(\mathbf{z}_{1:K}^c \mid \mathbf{x}, \mathbf{z}_{1:K}^m) = \prod_{k=1}^{K} q_\phi\big(\mathbf{z}_k^c \mid \mathbf{x}, \mathbf{z}_{1:k}^m\big)\,, \tag{7}$$

with the dependence on $\mathbf{z}_{1:k-1}^m$ realised by an RNN $\mathrm{R}_\phi$. The RNN could, in principle, be shared with the prior, but we have not investigated this option. All conditional distributions are Gaussian. For GENESIS-S, the approximate posterior takes the form $q_\phi(\mathbf{z}_{1:K} \mid \mathbf{x}) = \prod_{k=1}^{K} q_\phi(\mathbf{z}_k \mid \mathbf{x}, \mathbf{z}_{1:k-1})$.

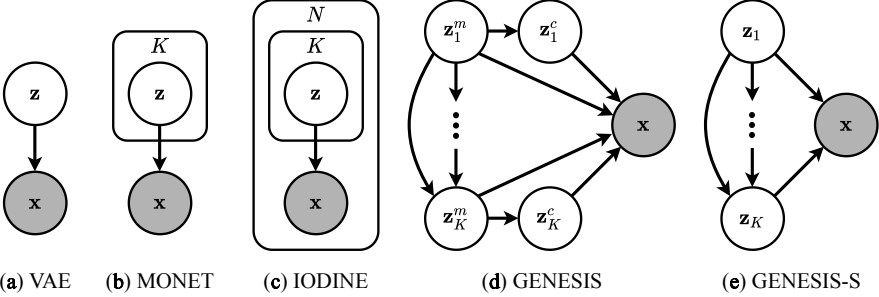

Figure 2: Graphical model of GENESIS compared to related methods. $N$ denotes the number of refinement iterations in IODINE. Unlike the other methods, both GENESIS variants explicitly model dependencies between scene components.

**Learning** GENESIS can be trained by maximising the evidence lower bound (ELBO) on the log-marginal likelihood $\log p_\theta(\mathbf{x})$, given by

$$\mathcal{L}_{\mathrm{ELBO}}(\mathbf{x}) = \mathbb{E}_{q_\phi(\mathbf{z}^c,\mathbf{z}^m|\mathbf{x})}\left[\log \frac{p_\theta(\mathbf{x} \mid \mathbf{z}^c,\mathbf{z}^m)p_\theta(\mathbf{z}^c \mid \mathbf{z}^m)p_\theta(\mathbf{z}^m)}{q_\phi(\mathbf{z}^c \mid \mathbf{z}^m,\mathbf{x})q_\phi(\mathbf{z}^m \mid \mathbf{x})}\right] \tag{8}$$

$$= \mathbb{E}_{q_\phi(\mathbf{z}^c,\mathbf{z}^m|\mathbf{x})}[\log p_\theta(\mathbf{x} \mid \mathbf{z}^c,\mathbf{z}^m)] - \mathrm{KL}\left(q_\phi(\mathbf{z}^c,\mathbf{z}^m \mid \mathbf{x}) \,\|\, p_\theta(\mathbf{z}^c,\mathbf{z}^m)\right) . \tag{9}$$

However, this often leads to a strong emphasis on the likelihood term, while allowing the marginal approximate posterior $q_\phi(\mathbf{z}) = \mathbb{E}_{p_{\mathrm{data}}(\mathbf{x})}[q_\phi(\mathbf{z} \mid \mathbf{x})]$ to drift away from the prior distribution, hence increasing the KL-divergence. This also decreases the quality of samples drawn from the model. To prevent this behaviour, we use the Generalised ELBO with Constrained Optimisation (GECO) objective from Rezende & Viola (2018) instead, which changes the learning problem to minimising the KL-divergence subject to a reconstruction constraint. Let $C \in \mathbb{R}$ be the minimum allowed reconstruction log-likelihood, GECO then uses Lagrange multipliers to solve the following problem,

$$\theta^\star, \phi^\star = \arg\min_{\theta,\phi} \mathrm{KL}\left(q_\phi(\mathbf{z}^c,\mathbf{z}^m \mid \mathbf{x}) \,\|\, p_\theta(\mathbf{z}^c,\mathbf{z}^m)\right)$$
$$\text{such that} \quad \mathbb{E}_{q_\phi(\mathbf{z}^c,\mathbf{z}^m|\mathbf{x})}[\log p_\theta(\mathbf{x} \mid \mathbf{z}^c,\mathbf{z}^m)] \geq C . \tag{10}$$

## 4 EXPERIMENTS

In this section, we present qualitative and quantitative results on *coloured Multi-dSprites* (Burgess et al., 2019), the "rooms-ring-camera" dataset from *GQN* (Eslami et al., 2018) and the *ShapeStacks* dataset (Groth et al., 2018). We use an image resolution of 64-by-64 for all experiments. The number of components is set to $K = 5$, $K = 7$, and $K = 9$ for Multi-dSprites, GQN, and ShapeStacks, respectively. More details about the datasets are provided in Appendix A. Implementation and training details of all models are described in Appendix B.

### 4.1 COMPONENT-WISE SCENE GENERATION

Unlike previous works, GENESIS has an autoregressive prior to capture intricate dependencies between scene components. Modelling these relationships is necessary to generate coherent scenes. For example, different parts of the background need to fit together; we do not want to create components such as the sky several times; and several physical objects cannot be in the same location. GENESIS is able to generate novel scenes by sequentially sampling scene components from the prior and conditioning each new component on those that have been generated during previous steps.

After training GENESIS and MONet on the GQN dataset, Figure 3 shows the component-by-component generation process of novel scenes, corresponding to drawing samples from the respective prior distributions. More examples of generated scenes are shown in Figure 6, Appendix D. With GENESIS, either an object in the foreground or a part of the background is generated at every step and these components fit together to make up a semantically consistent scene that looks similar to the training data. MONet, though, generates random artefacts at every step that do not form a sensible scene. These results are striking but not surprising: MONet was not designed for scene generation. The need for such a model is why we developed GENESIS.

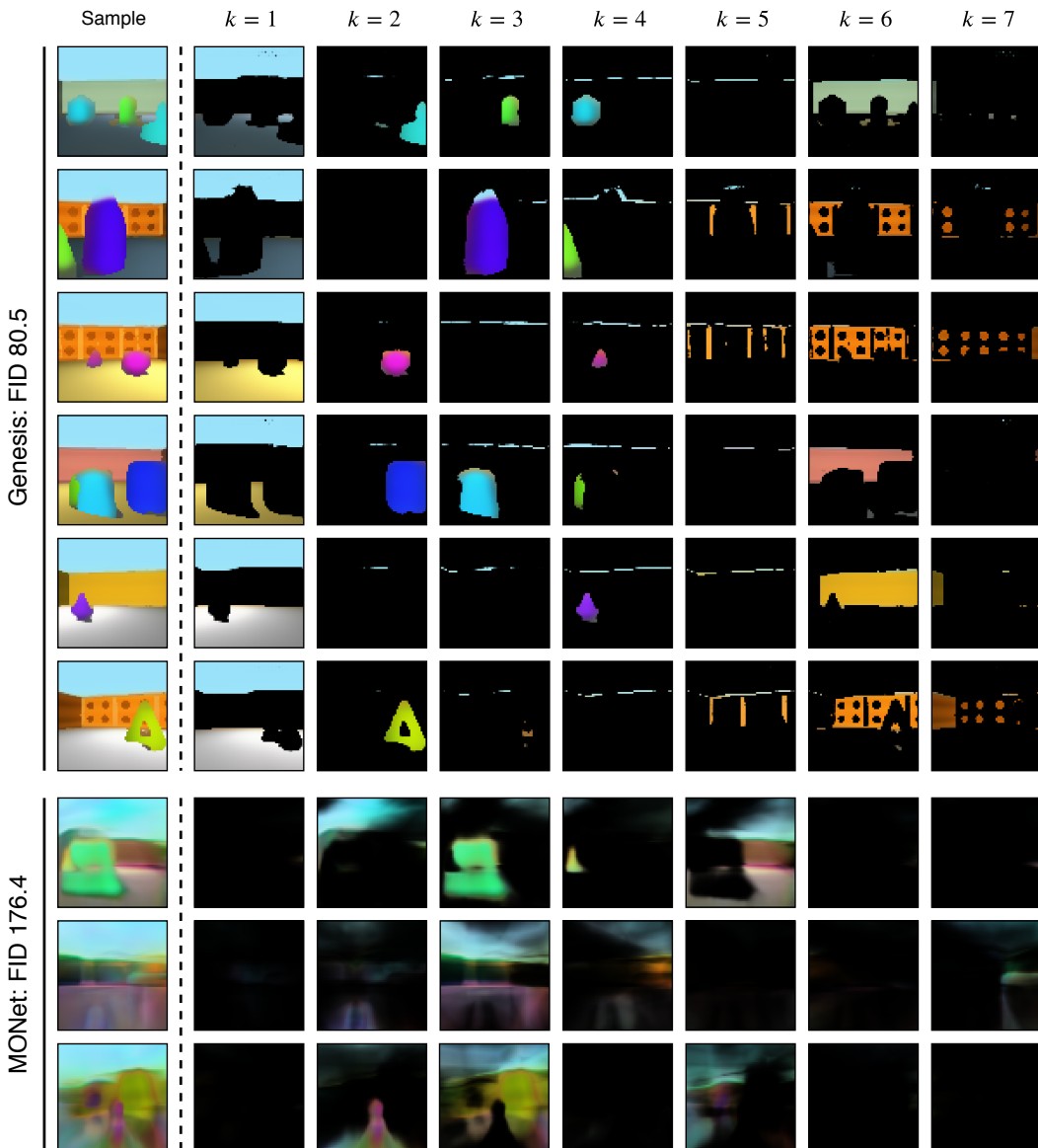

Figure 3: Component-by-component scene generation with GENESIS and MONet after training on the GQN dataset. The first pane shows the final scene and the subsequent panes show the components generated at each step. GENESIS first generates the sky and the floor, followed by individual objects, and finally distinct parts of the wall in the background to compose a coherent scene. MONet, in contrast, only generates incomplete components that do not fit together.

Notably, GENESIS pursues a consistent strategy for scene generation: Step one generates the floor and the sky, defining the layout of the scene. Steps two to four generate individual foreground objects. Some of these slots remain empty if less than three objects are present in the scene. The final three steps generate the walls in the background. We conjecture that this strategy evolves during training as the floor and sky constitute large and easy to model surfaces that have a strong impact on the reconstruction loss. Finally, we observe that some slots contain artefacts of the sky at the top of the wall boundaries. We conjecture this is due to the fact that the mask decoder does not have skip connections as typically used in segmentation networks, making it difficult for the model to predict sharp segmentation boundaries. Scenes generated by GENESIS-S are shown in Figure 8 and Figure 9, Appendix D. While GENESIS-S does separate the foreground objects from the background, it generates them in one step and the individual background components are not very interpretable.

## 4.2 INFERENCE OF SCENE COMPONENTS

Like MONet and IODINE, which were designed for unsupervised scene decomposition, GENESIS is also able to segment scenes into meaningful components. Figure 4 compares the decomposition of two images from the GQN dataset with GENESIS and MONet. Both models follow a similar decomposition strategy, but MONet fails to disambiguate one foreground object in the first example and does not reconstruct the background in as much detail in the second example. In Appendix E, Figure 10 illustrates the ability of both methods to disambiguate objects of the same colour and Figure 11 shows scene decomposition with GENESIS-S.

Following Greff et al. (2019), we quantify segmentation performance with the Adjusted Rand Index (ARI) of pixels overlapping with ground truth foreground objects. We computed the ARI on 300 random images from the ShapeStacks test set for five models trained with different random seeds. GENESIS achieves an ARI of $0.73 \pm 0.03$ which is better than $0.63 \pm 0.07$ for MONet. This metric, however, does not penalise objects being over-segmented, which can give a misleading impression with regards to segmentation quality. This is illustrated in Figure 13, Appendix E.

Inspired by Arbelaez et al. (2010), we thus propose to use the *segmentation covering* (SC) of the ground truth foreground objects by the predicted masks. This involves taking a weighted mean over mask pairs, putting a potentially undesirable emphasis on larger objects. We therefore also consider taking an unweighted mean (mSC). For the same 300 images from the ShapeStacks test set and five different random seeds, GENESIS (SC: $0.64 \pm 0.08$, mSC: $0.60 \pm 0.09$) again outperforms MONet (SC: $0.52 \pm 0.09$, mSC: $0.49 \pm 0.09$). More details are provided in Appendix C.

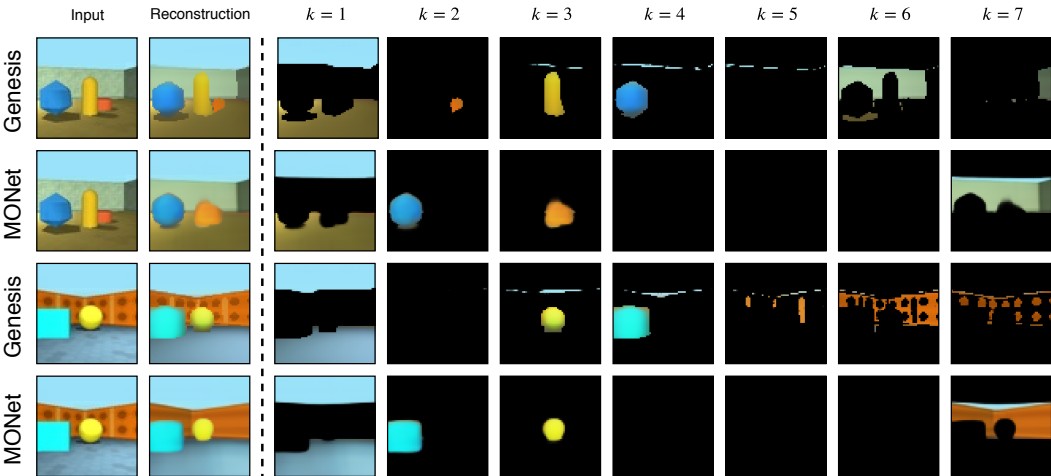

Figure 4: Step-by-step decomposition of the same scene from GQN with GENESIS and MONet. Unlike MONet, GENESIS clearly differentiates individual objects in the first example. In the second example, GENESIS captures the fine-grained pattern of the wall in the background better than MONet.

## 4.3 EVALUATION OF UNSUPERVISED REPRESENTATION UTILITY

Using a subset of the available labelled training images from ShapeStacks, we train a set of classifiers on the representations learned by GENESIS and several baselines to evaluate how well these representations capture the ground truth scene state. In particular, we consider three tasks: (1) Is a tower stable or not? (2) What is the tower's height in terms of the number of blocks? (3) What is the camera viewpoint (out of 16 possibilities)? Tower stability is a particularly interesting property as it depends on in fine-grained object information and the relative positioning of objects. We selected the third task as learning scene representations from different views has previously been prominently explored in Eslami et al. (2018). We compare GENESIS and GENESIS-S against three baselines: MONet, a VAE with a spatial broadcast decoder (BD-VAE) and a VAE with a deconvolutional decoder (DC-VAE). The results are summarised in Table 1. The architectural details of the baselines are described in Appendix B.2 and Appendix B.3. The implementation details of the classifiers are provided in Appendix B.5.

Both GENESIS and GENESIS-S perform better than than the baselines at predicting tower stability and their accuracies on predicting the height of the towers is only outperformed by MONet. We conjecture that MONet benefits here by its deterministic segmentation network. Overall, this corroborates the intuition that object-centric representations are indeed beneficial for these tasks which focus on the foreground objects. We observe that the BD-VAE does better than the DC-VAE on all three tasks, reflecting the motivation behind its design which is aimed at better disentangling the underlying factors of variation in the data (Watters et al., 2019b). All models achieve a high accuracy at predicting the camera view. Finally, we note that none of models reach the stability prediction accuracies reported in Groth et al. (2018) which were obtained with an Inception-v4 classifier (Szegedy et al., 2017). This is not surprising considering that only a subset the training images is used for training the classifiers without data augmentation and at a reduced resolution.

Table 1: Classification accuracy in % on the test sets of the ShapeStacks tasks.

| Task | GENESIS | GENESIS-S | MONet | BD-VAE | DC-VAE | Random |
|---|---|---|---|---|---|---|
| Stability | **64.0** | 63.2 | 59.6 | 60.1 | 59.0 | 50.0 |
| Height | 80.3 | 80.8 | **88.4** | 78.6 | 67.5 | 22.8 |
| View | 99.3 | **99.7** | 99.5 | **99.7** | 99.1 | 6.25 |

## 4.4 QUANTIFYING SAMPLE QUALITY

In order to quantify the quality of generated scenes, Table 2 summarises the Fréchet Inception Distances (FIDs) (Heusel et al., 2017) between 10,000 images generated by GENESIS as well several baselines and 10,000 images from the Multi-dSprites and the GQN test sets, respectively. The two GENESIS variants achieve the best FID on both datasets. While GENESIS-S performs better than GENESIS on GQN, Figure 8 and Figure 9 in Appendix D show that individual scene components are less interpretable and that intricate background patterns are generated at the expense of sensible foreground objects. It is not surprising that the FIDs for MONet are relatively large given that it was not designed for generating scenes. Interestingly, the DC-VAE achieves a smaller FID on GQN than the BD-VAE. This is surprising given that the BD-VAE representations are more useful for the ShapeStacks classification tasks. Given that the GQN dataset and ShapeStacks are somewhat similar in structure and appearance, this indicates that while FID correlates with perceptual similarity, it does not necessarily correlate with the general utility of the learned representations for downstream tasks. We include scenes sampled from the BD-VAE and the DC-VAE in Figure 7, Appendix D, where we observe that the DC-VAE models the background fairly well while foreground objects are blurry.

Table 2: Fréchet Inception Distances for GENESIS and baselines on GQN.

| Dataset | GENESIS | GENESIS-S | MONet | BD-VAE | DC-VAE |
|---|---|---|---|---|---|
| Multi-dSprites | **24.9** | 28.2 | 92.7 | 89.8 | 100.5 |
| GQN | 80.5 | **70.2** | 176.4 | 145.5 | 82.5 |

## 5 CONCLUSIONS

In this work, we propose a novel object-centric latent variable model of scenes called GENESIS. We show that GENESIS is, to the best of our knowledge, the first unsupervised model to both decompose rendered 3D scenes into semantically meaningful constituent parts, while at the same time being able to generate coherent scenes in a component-wise fashion. This is achieved by capturing relationships between scene components with an autoregressive prior that is learned alongside a computationally efficient sequential inference network, setting GENESIS apart from prior art. Regarding future work, an interesting challenge is to scale GENESIS to more complex datasets and to employ the model in robotics or reinforcement learning applications. To this end, it will be necessary to improve reconstruction and sample quality, reduce computational cost, and to scale the model to higher resolution images. Another potentially promising research direction is to adapt the formulation to only model parts of the scene that are relevant for a certain task.

ACKNOWLEDGMENTS

This research was supported by an EPSRC Programme Grant (EP/M019918/1), an EPSRC DTA studentship, and a Google studentship. The authors would like to acknowledge the use of the University of Oxford Advanced Research Computing (ARC) facility in carrying out this work, `http://dx.doi.org/10.5281/zenodo.22558`, and the use of Hartree Centre resources. The authors would like to thank Yizhe Wu for his help with re-implementing MONeT, Oliver Groth for his support with the GQN and ShapeStacks datasets, and Rob Weston for proof reading the paper.

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

# A  DATASETS

**Multi-dSprites (Burgess et al., 2019)**  Images contain between one and four randomly selected "sprites" from Matthey et al. (2017), available at `https://github.com/deepmind/dsprites-dataset`. For each object and the background, we randomly select one of five different, equally spread values for each of the three colour channels and generate 70,000 images. We set aside 10,000 for validation and testing each. The script for generating this data will be released with the rest of our code.

**GQN (Eslami et al., 2018)**  The "rooms-ring-camera" dataset includes simulated 3D scenes of a square room with different floor and wall textures, containing one to three objects of various shapes and sizes. It can be downloaded from `https://github.com/deepmind/gqn-datasets`.

**ShapeStacks (Groth et al., 2018)**  Images show simulated block towers of different heights (two to six blocks). Individual blocks can have different shapes, sizes, and colours. Scenes have annotations for: stability of the tower (binary), number of blocks (two to six), properties of individual blocks, locations in the tower of centre-of-mass violations and planar surface violations, wall and floor textures (five each), light presets (five), and camera view points (sixteen). More details about the dataset and download links can be found at `https://shapestacks.robots.ox.ac.uk/`.

# B  IMPLEMENTATION DETAILS

## B.1  GENESIS ARCHITECTURE

We use the architecture from Berg et al. (2018) to encode and decode $\mathbf{z}_k^m$ with the only modification of applying batch normalisation (Ioffe & Szegedy, 2015) before the GLU non-linearities (Dauphin et al., 2017). The convolutional layers in the encoder and decoder have five layers with size-5 kernels, strides of [1, 2, 1, 2, 1], and filter sizes of [32, 32, 64, 64, 64] and [64, 32, 32, 32, 32], respectively. Fully-connected layers are used at the lowest resolution.

The encoded image is passed to a long short-term memory (LSTM) cell (Hochreiter & Schmidhuber, 1997) followed by a linear layer to compute the mask latents $\mathbf{z}_k^m$ of size 64. The LSTM state size is twice the latent size. Importantly, unlike the analogous counterpart in MONet, the decoding of $\mathbf{z}_k^m$ is performed in parallel. The autoregressive prior $p_\theta\big(\mathbf{z}_k^m \mid \mathbf{z}_{1:k-1}^m\big)$ is implemented as an LSTM with 256 units. The conditional distribution $p_\theta(\mathbf{z}_k^c \mid \mathbf{z}_k^m)$ is parameterised by a multilayer perceptron (MLP) with two hidden layers, 256 units per layer, and ELUs (Clevert et al., 2016). We use the same component VAE featuring a spatial broadcast decoder as MONet to encode and decode $z_k^c$, but we replace RELUs (Glorot et al., 2011) with ELUs.

For GENESIS-S, as illustrated in Figure 5, the encoder of $\mathbf{z}_k$ is the same as for $\mathbf{z}_k^m$ above and the decoder from Berg et al. (2018) is again used to compute the mixing probabilities. However, GENESIS-S also has a second decoder with spatial broadcasting to obtain the scene components $\mathbf{x}_k$ from $\mathbf{z}_k$. We found the use of two different decoders to be important for GENESIS-S in order for the model to decompose the input.

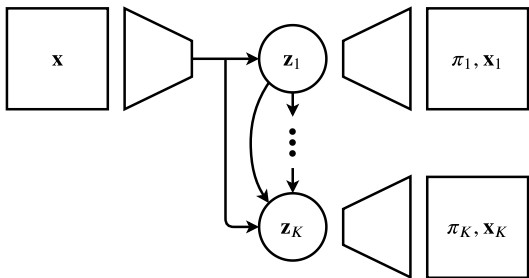

Figure 5: GENESIS-S overview. Given an image $\mathbf{x}$, an encoder and an RNN compute latent variables $\mathbf{z}_k$. These are decoded to directly obtain the mixing probabilities $\pi_k$ and the scene components $\mathbf{x}_k$.

## B.2 MONet Baselines

We followed the provided architectural details described in Burgess et al. (2019). Regarding unspecified details, we employ an attention network with [32, 32, 64, 64, 64] filters in the encoder and the reverse in the decoder. Furthermore, we normalise the mask prior with a softmax function to compute the KL-divergence between mask posterior and prior distributions.

## B.3 VAE Baselines

Both the BD-VAE and the DC-VAE have a latent dimensionality of 64 and the same encoder as in Berg et al. (2018). The DC-VAE also uses the decoder from Berg et al. (2018). The BD-VAE has the same spatial broadcast decoder with ELUs as GENESIS, but with twice the number of filters to enable a better comparison.

## B.4 Optimisation

The scalar standard deviation of the Gaussian image likelihood components is set to $\sigma_x = 0.7$. We use GECO (Rezende & Viola, 2018) to balance the reconstruction and KL divergence terms in the loss function. The goal for the reconstruction error is set to $0.5655$, multiplied by the image dimensions and number of colour channels. We deliberately choose a comparatively weak reconstruction constraint for the GECO objective to emphasise KL minimisation and sample quality. For the remaining GECO hyperparameters, the default value of $\alpha = 0.99$ is used and the step size for updating $\beta$ is set to $10^{-5}$. We increase the step size to $10^{-4}$ when the reconstruction constraint is satisfied to accelerate optimisation as $\beta$ tended to undershoot at the beginning of training.

All models are trained for $5 * 10^5$ iterations with a batch size of 32 using the ADAM optimiser (Kingma & Ba, 2015) and a learning rate of $10^{-4}$. With these settings, training GENESIS takes about two days on a single GPU. However, we expect performance to improve with further training. This particularly extends to training GENESIS on ShapeStacks where $5 * 10^5$ training iterations are not enough to achieve good sample quality.

## B.5 ShapeStacks Classifiers

Multilayer perceptrons (MLPs) with one hidden layer, 512 units, and ELU activations are used for classification. The classifiers are trained for 100 epochs on 50,000 labelled examples with a batch size of 128 using a cross-entropy loss, the ADAM optimiser, and a learning rate of $10^{-4}$. As inputs to the classifiers, we concatenate $\mathbf{z}_k^m$ and $\mathbf{z}_k^c$ for GENESIS, $\mathbf{z}_k$ for GENESIS-S, and the component VAE latents for the two MONet variants.

## C  Segmentation Covering

Following Arbelaez et al. (2010), the *segmentation covering* (SC) is based on the intersection over union (IOU) between pairs of segmentation masks from two sets $S$ and $S'$. In this work, we consider $S$ to be the segmentation masks of the ground truth foreground objects and $S'$ to be the predicted segmentation masks. The covering of $S$ by $S'$ is defined as:

$$C(S' \to S) = \frac{1}{\sum_{R \in S} |R|} \sum_{R \in S} |R| \max_{R' \in S'} \text{IOU}(R, R'), \tag{11}$$

where $|R|$ denotes the number of pixels belonging to mask $R$. Note that this formulation is slightly more general than the one in Arbelaez et al. (2010) which assumes that masks in $S$ are non-overlapping and cover the entire image. The above takes a weighted mean over IOU values, proportional to the number of pixels of the masks being covered. To give equal importance to masks of different sizes, we also consider taking an unweighted mean (mSC):

$$C_m(S' \to S) = \frac{1}{|S|} \sum_{R \in S} \max_{R' \in S'} \text{IOU}(R, R'), \tag{12}$$

where $|S|$ denotes the number of non-empty masks in $S$. Importantly and unlike the ARI, both segmentation covering variations penalise the over-segmentation of ground truth objects as this decreases the IOU for a pair of masks. This is illustrated in Figure 13, Appendix E.

# D    COMPONENT-WISE SCENE GENERATION - GQN

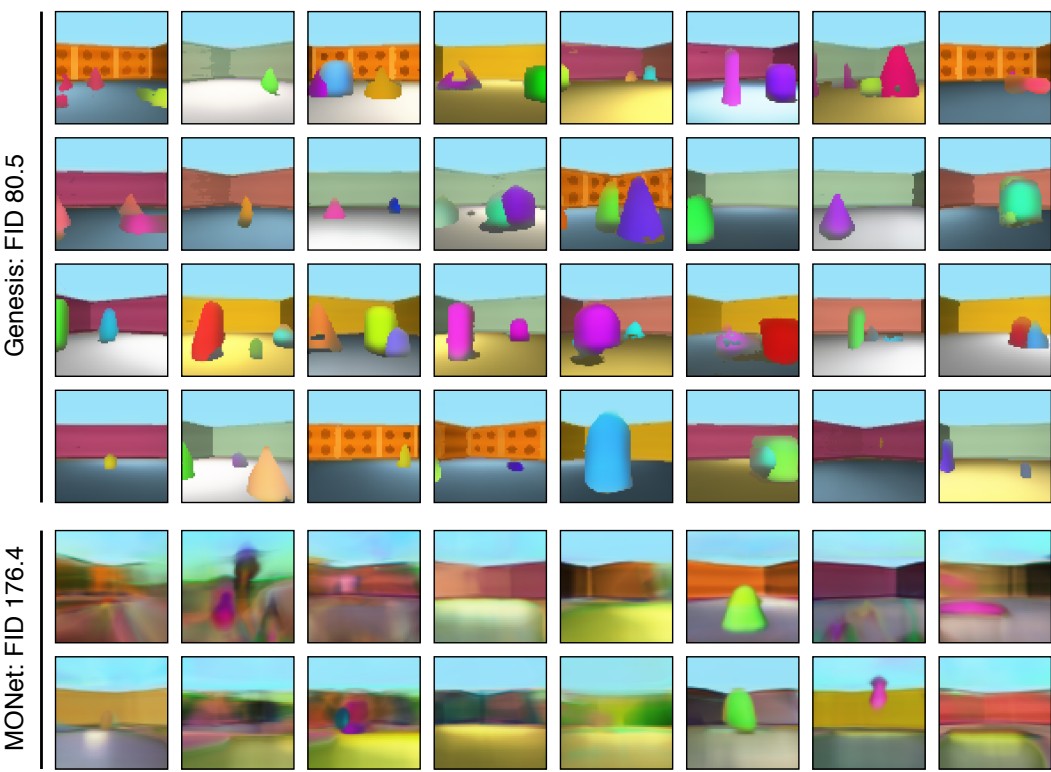

Figure 6: Randomly selected scenes generated by GENESIS and MONET after training on the GQN dataset. Images sampled from GENESIS contain clearly distinguishable foreground objects and backgrounds. Samples from MONET, however, are mostly incoherent.

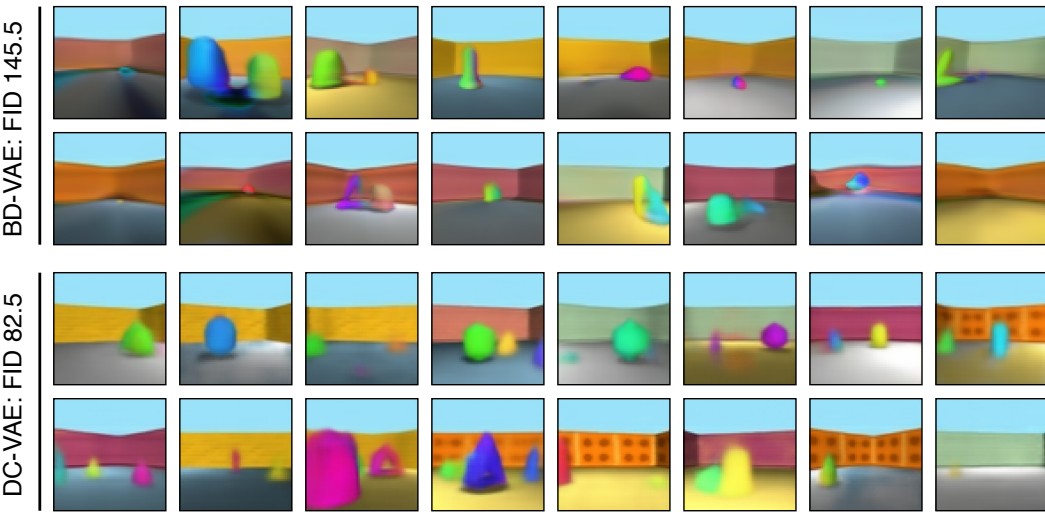

Figure 7: Randomly selected scenes generated by the BD-VAE and the DC-VAE after training on the GQN dataset; shown for comparison. The DC-VAE generates decent scene backgrounds but foreground objects are blurry.

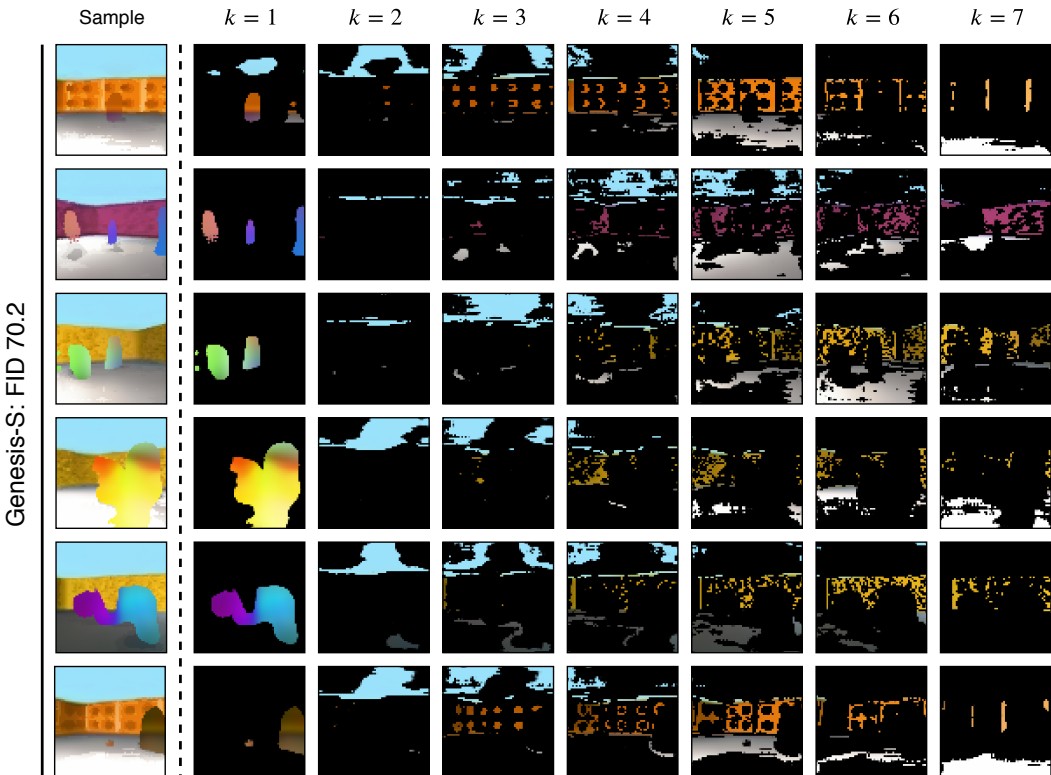

Figure 8: Component-by-component scene generation with GENESIS-S after training on the GQN dataset. While GENESIS-S nominally achieves the best FID in Table 2, this appears to be due to the generation of high fidelity background patterns rather than appropriate foreground objects. Furthermore, unlike the components generated by GENESIS at every step in Figure 3, the components generated by GENESIS-S are not very interpretable.

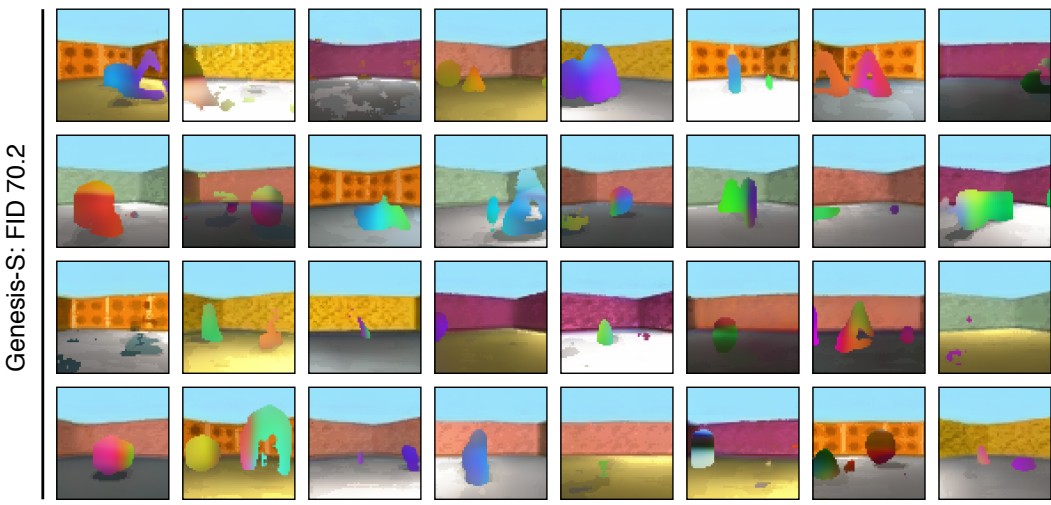

Figure 9: Randomly selected scenes generated by GENESIS-S after training on the GQN dataset.

# E    INFERENCE OF SCENE COMPONENTS

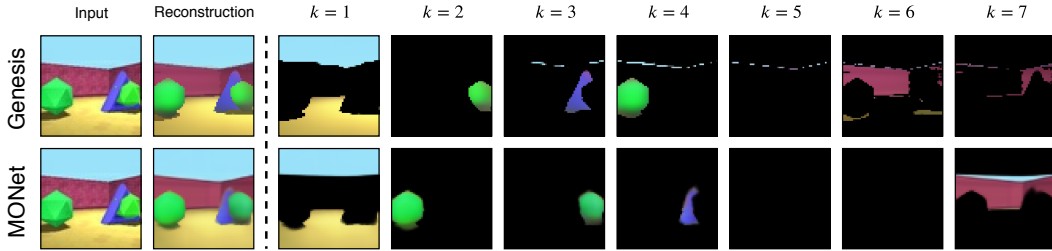

Figure 10: Step-by-step decomposition of a scene from GQN with GENESIS and MONet. Two objects with the same shape and colour are successfully identified by both models. While colour and texture are useful cues for decomposition, this example shows that both models perform something more useful than merely identifying regions of similar colour.

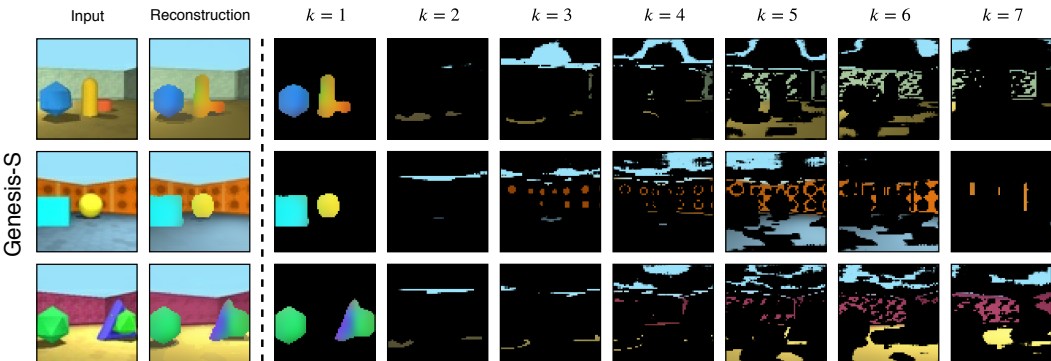

Figure 11: Step-by-step decomposition of the same scenes as in Figure 4 and Figure 10 with GENESIS-S. While the foreground objects are distinguished from the background, they are explained together in the first step. Subsequent steps reconstruct the background in a haphazard fashion.

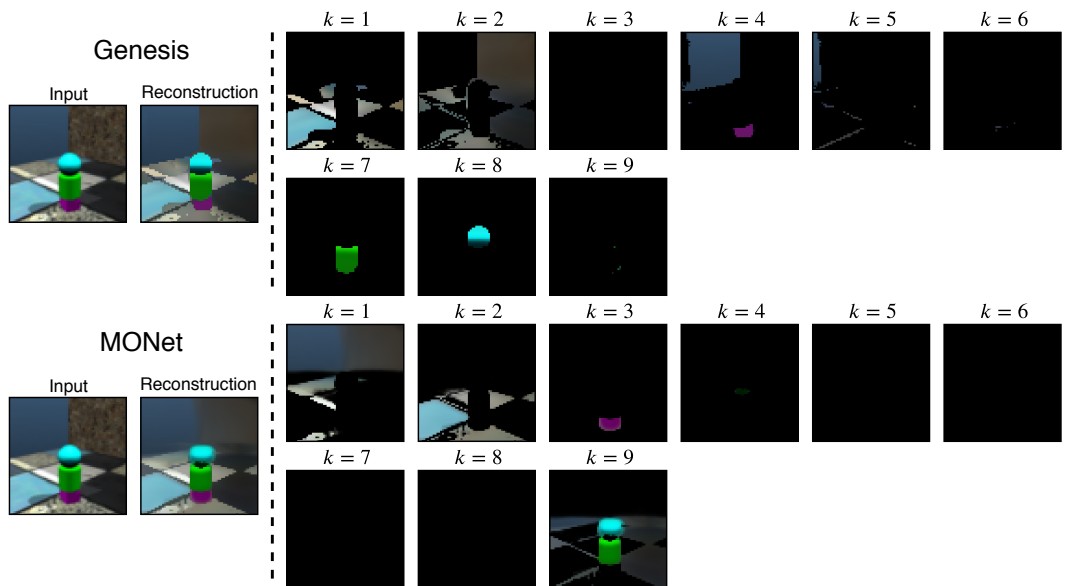

Figure 12: A ShapeStacks tower is decomposed by GENESIS and MONet. Compared to the GQN dataset, both methods struggle to segment the foreground objects properly. GENESIS captures the purple shape and parts of the background wall in step $k = 4$. MONet explains the green shape, the cyan shape, and parts of floor in step $k = 9$. This is reflected in the foreground ARI and segmentation covering for GENESIS (ARI: 0.82, SC: 0.68, mSC: 0.58) and MONet (ARI: 0.39, SC: 0.26, mSC: 0.35); the latter being lower as the green and cyan shapes are not separated.

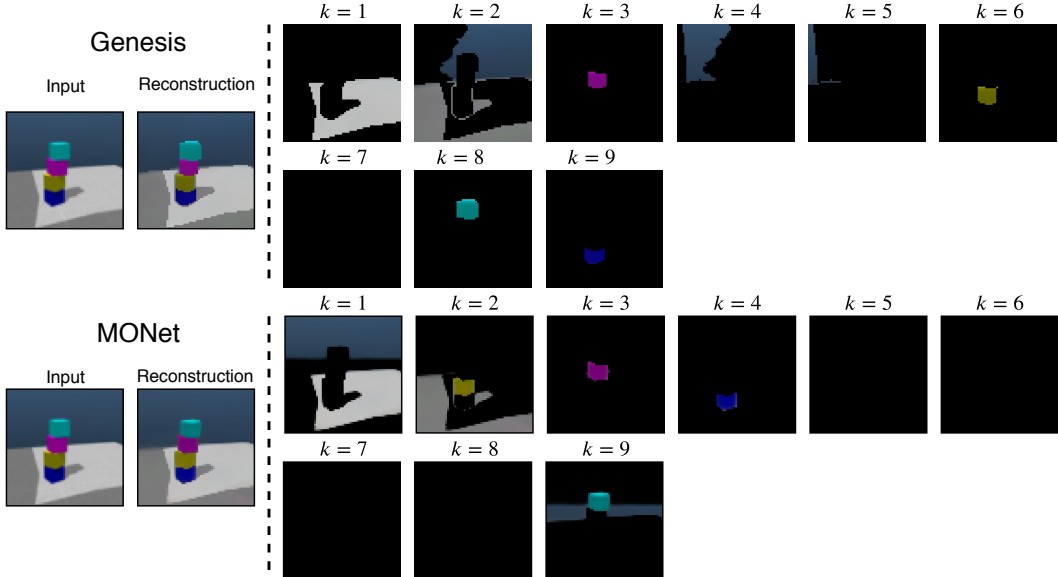

Figure 13: In this example, GENESIS (ARI: 0.83, SC: 0.83, mSC: 0.83) segments the four foreground objects properly. MONet (ARI: 0.89, SC: 0.47, mSC: 0.50), however, merges foreground objects and background again in steps $k = 2$ and $k = 9$. Despite the inferior decomposition, the ARI for MONet is higher than for GENESIS. This is possible as the ARI does not penalise the over-segmentation of the foreground objects, highlighting its limitations for evaluating unsupervised instance segmentation. The segmentation covering, however, reflects the quality of the segmentatioin masks properly.

