# OpenReview forum: "GENESIS: Generative Scene Inference and Sampling with Object-Centric Latent Representations"
_ICLR.cc/2020/Conference — Accept (Poster)_

### Official Review · AnonReviewer3 · 2019-10-22
**Official Blind Review #3**

**Rating:** 8

**Review:**

UPDATE: I appreciated the authors' discussion. The authors addressed my questions satisfactorily, and I maintain my original rating of accept.

----
Summary: This papers tackles the question of building an object-centric latent variable generative model of scenes that can sample novel scenes with coherent objects and relationships. To do this, the authors define a generative model, GENESIS, that uses an autoregressive prior over mask variables. The component variables are generated conditioned on these mask variables. The visual appearance of the objects are generated conditioned on the component variables. Inference is done sequentially by inferring some later object varaibles conditioned on others. Results show that the model adopts a consistent strategy in generating and inferring the scene components, first considering the background then the foreground objects. The authors apply GENESIS to three datasets with monochromatic objects and show that GENESIS qualitatively generates coherent scenes and infers coherent scene components.

Decision: Accept. This work clearly addresses a problem beyond current object-centric modeling approaches such as IODINE and MONet, which is the problem of generating novel scenes.

Strengths:
- The paper is well written and executed.
- The evaluation is thorough
- The problem and solution are well motivated

Weaknesses:
- While the authors demonstrates that GENESIS is able to model static scenes, it is not clear how straightforward it is to extend GENESIS to modeling dynamics for the purpose of robotics and reinforcement learning (as stated in the authors' motivation). Whereas approaches such as IODINE or RNEM (van Steenkiste et al. 2018) treat the object latent that can be propagated through time, maintatining that the same latent models the same object may not be a guarantee for autoregressive approaches such as MONet or GENESIS that re-parse the scene at every frame. Object temporal consistency is especially important when considering tasks that have occlusion, which are important problems in robotics.
- The results in Appendix D seem to suggest that GENESIS decomposes a scene mostly via color segmentation, as IODINE and MONet do. One concern is that such models that rely mainly on color segmentation are not applicable for real world robotics with various lighting conditions and textures because segmenting based on color may not provide coherent object representations. Would the authors be able to provide an empirical analysis of how GENESIS models a real-world scene, analogous to Figure 11 in the IODINE paper?

Van Steenkiste, S., Chang, M., Greff, K., & Schmidhuber, J. (2018). Relational neural expectation maximization: Unsupervised discovery of objects and their interactions. arXiv preprint arXiv:1802.10353.

**Experience Assessment:**

I have published one or two papers in this area.

**Review Assessment: Checking Correctness Of Derivations And Theory:**

I assessed the sensibility of the derivations and theory.

**Review Assessment: Checking Correctness Of Experiments:**

I assessed the sensibility of the experiments.

**Review Assessment: Thoroughness In Paper Reading:**

I read the paper thoroughly.

---

> ### Author Response · Authors · 2019-11-09
> **Reply to reviewer 3 from the authors**
>
> We thank the reviewer for the positive feedback. Below we comment on the two identified weaknesses.
>
> Reviewer:
> “While the authors demonstrates that GENESIS is able to model static scenes, it is not clear how straightforward it is to extend GENESIS to modeling dynamics for the purpose of robotics and reinforcement learning (as stated in the authors' motivation).”
> Answer:
> We agree that temporal consistency is important in perception. IODINE models dynamic scenes by maintaining a state between time steps and updating the posterior statistics in an additive fashion. It should be possible to modify GENESIS to follow the same procedure: the posterior from step t-1 can be used as an additional input and/or prior when inferring an additive update to the posterior at step t. This would correspond to replacing IODINE’s encoder and decoder with GENESIS, while using the same state transition mechanism.
>
> Reviewer:
> “The results in Appendix D seem to suggest that GENESIS decomposes a scene mostly via color segmentation, as IODINE and MONet do. One concern is that such models that rely mainly on color segmentation are not applicable for real world robotics with various lighting conditions and textures because segmenting based on color may not provide coherent object representations. Would the authors be able to provide an empirical analysis of how GENESIS models a real-world scene, analogous to Figure 11 in the IODINE paper?”
> Answer:
> We agree that current methods are not suitable yet for real world applications and we are looking forward to future work on this. Temporal information should be a useful cue to overcome this limitation. Preliminary experiments indicated that like IODINE, GENESIS also appears to segment natural images into regions of similar colour rather than meaningful objects. We are going to further investigate this. Nevertheless, we believe that by providing a fully probabilistic model of scenes that captures relationships between components and a more scalable, probabilistic attention mechanism, GENESIS overcomes key limitations of previous works and therefore makes valuable contributions towards this goal.

---

> > ### Comment · AnonReviewer3 · 2019-11-14
> > **Reply**
> >
> > With regard to temporal consistency, how would GENESIS guarantee that the indices of the latents of the posterior at timestep t correspond to the indices of the latents of the posterior at timestep t-1? This would be important, for example, when objects are occluded and reappear at a later timestep. It may be possible to apply a single step transition model to the latents inferred at timestep t-1 to predict the latents at timestep t does, as COBRA (Watters et al. 2019) does, but it is not clear how to update the predicted latents at timestep t with new information from the observation of timestep t, using this method, because this would require finding a correspondence between the latents outputted by the encoder with the latents outputted by the transition model. One way to establish such a correspondence was recently shown in OP3 (Veerapaneni et al. 2019), which models the latents symmetrically as IODINE does. But to simply replace IODINE's encoder and decoder with an autoregressive one such as GENESIS seems rather nontrivial and may require an entirely new inference algorithm. Therefore, while GENESIS seems to make sense as a model for static scenes, it's still not clear to me how straightforward it is to extend GENESIS to modeling dynamics for the purpose of robotics and reinforcement learning.
> >
> > Watters, N., Matthey, L., Bosnjak, M., Burgess, C. P., & Lerchner, A. (2019). COBRA: Data-Efficient Model-Based RL through Unsupervised Object Discovery and Curiosity-Driven Exploration. arXiv preprint arXiv:1905.09275.
> >
> > Veerapaneni, R., Co-Reyes, J. D., Chang, M., Janner, M., Finn, C., Wu, J., ... & Levine, S. (2019). Entity Abstraction in Visual Model-Based Reinforcement Learning. arXiv preprint arXiv:1910.12827.

---

> > > ### Author Response · Authors · 2019-11-15
> > > **Reply from the authors**
> > >
> > > Neither IODINE nor OP3 provide a formal guarantee that object indices are consistent across time steps. This is induced by the model and the inference mechanism, in particular the use of “auxiliary variables” to break symmetry. Unlike these methods, GENESIS infers objects sequentially and therefore imposes an ordering which follows a fairly consistent, learned strategy.
> > >
> > > Thus, we would expect that for small changes in scene state between time steps t-1 and t, the ordering will be very similar (or potentially identical) to the previous time step, even when running the model only on a frame-by-frame basis. Furthermore, we could pass the statistics of the previous posterior for object k, i.e. lambda^{t-1}_k = {mu^{t-1}_k, sigma^{t-1}_k}, as an additional input only at inference step k when inferring lambda^{t}_k, e.g. by updating the approximate posterior in an additive fashion (similar to IODINE):
> > >
> > > lambda^{t}_k = lambda^{t-1}_k + Linear(LSTM([lambda^{t-1}_k, e^{t}, e_{<t}], h_{<k}))
> > >
> > > where e^{t} is an encoding of the current image, e_{<t} is a (possibly optional) encoding capturing image history, h_{<k} is the LSTM hidden state, and [.] denotes concatenation; should be a further strong inductive bias for the model to maintain the same object ordering. SQAIR [1] follows a similar strategy, but it uses a sample from the approximate posterior instead of its statistics as an auxiliary input.
> > >
> > > Instead of an additive update, another option would indeed be to move further away from IODINE by learning a transition model from the state at t-1 to obtain a prior which is combined with a data-dependant “state-likelihood term” to arrive at a new posterior at time step t via Bayes' rule, which can be done analytically if all distributions are Gaussians.
> > >
> > > We admit that trying to enforce a fixed ordering could lead to other issues and that it is difficult to tell whether this would work well without trying it. We very much appreciate the reviewer’s thoughts and find this an interesting discussion.
> > >
> > > [1] Kosiorek et al. "Sequential Attend, Infer, Repeat: Generative Modelling of Moving Objects." NeurIPS (2018).

---

### Official Review · AnonReviewer1 · 2019-10-23
**Official Blind Review #1**

**Rating:** 6

**Review:**

The authors propose a probabilistic generative latent variable model representing a 2D image as a mixture of latent components. It formulates the scene generation problem as a spatial Gaussian mixture model where each Gaussian component comes from the decoding of an object-centric latent variable. The contribution of the proposed method from previous works is the introduction of an autoregressive prior on the component latents. This allows the model to capture autoregressive dependencies among different components and thus help generate coherent scenes, which has not been shown in the previous works. In the experiments, the authors compare GENESIS with MONet and VAEs qualitatively and quantitatively and show that the model outperforms the baseline in terms of both scene decomposition and generation.

The proposed model seems like the right direction to improve upon MONet and it is nice to see the generation results. It is also nice to see the fully probabilistic modeling of the problem. Although it would improve over the MONet, I'm nevertheless not sure if the framework of sequential component generation (applying both MONet and GENESIS) can be a robust approach to more complex scenes, e.g., with a larger number of objects. Also, the mixture-based approach seems not guarantee the object-level decomposition. For example, if in the scene there are many objects of the same shape, size, and color, I think the proposed model may not properly distinguish them, as shown in some of the wall patterns in the experiments. But, I'm not sure what would happen if K is set to a large number to deal with this. Then, it would have the problem of long-term dependency in sequences.

Some comments and questions:
1. It would be good to show the qualitative result of the simplified Genesis-s and its qualitative result in the FID experiment.
2. Is the VAE baselines (BD-VAE and DC-VAE) trained by maximizing the ELBO? If it is the case, will the result of the FID experiment section be different if we train the VAE baseline with GECO?
3. Some detail of the implementation is missing, e.g. how do we model the posterior q_{\phi}(z_{k}^{c} | x, z_{1: k}^{m}).


**Experience Assessment:**

I have published one or two papers in this area.

**Review Assessment: Checking Correctness Of Derivations And Theory:**

I carefully checked the derivations and theory.

**Review Assessment: Checking Correctness Of Experiments:**

I carefully checked the experiments.

**Review Assessment: Thoroughness In Paper Reading:**

I read the paper thoroughly.

---

> ### Author Response · Authors · 2019-11-09
> **Reply to reviewer 1 from the authors**
>
> We thank the reviewer for the useful feedback. Below we attempt to address the comments and questions.
>
> Reviewer:
> “The contribution of the proposed method from previous works is the introduction of an autoregressive prior on the component latents.”
> Answer:
> The main contribution of our work is an unsupervised, object-centric generative model of rendered 3D scenes capable of both decomposing and generating scenes. One mechanism to achieve this is indeed the autoregressive prior. Additional contributions that differentiate GENESIS from prior works include a more efficient parameterisation of the stick-breaking attention process from MONet as well as a fully probabilistic inference mechanism for spatial GMMs (in contrast to the deterministic mechanism in MONet) that does not rely on the computationally expensive iterative refinement used in TAGGER, NEM, R-NEM, and IODINE. Finally, we also hope that the release of our code (implementing both GENESIS and MONET) will be useful for the community.
>
> Reviewer:
> “Although it would improve over the MONet, I'm nevertheless not sure if the framework of sequential component generation (applying both MONet and GENESIS) can be a robust approach to more complex scenes, e.g., with a larger number of objects.
> Answer:
> We would like to note that scenes of practical interest have a limited number of components. For example, the NYU Depth V2 dataset contains labels for a long tail of objects and the largest number of object instances per scene is around 50. LSTMs (as used in GENESIS) and other sequential models have been successfully applied to much longer sequences in large-scale applications. Crawford and Pineau present a model that performs sequential inference on Multi-MNIST, SET, and ATARI games with a very large number of components in "Spatially Invariant Unsupervised Object Detection with Convolutional Neural Networks" (AAAI 2019). This indicates that sequential inference of a larger number of scene components should in principle also be feasible for more complex datasets. We added this reference to the related work section.
>
> Reviewer:
> “Also, the mixture-based approach seems not guarantee the object-level decomposition. For example, if in the scene there are many objects of the same shape, size, and color, I think the proposed model may not properly distinguish them, as shown in some of the wall patterns in the experiments.”
> Answer:
> We agree that a mixture-model alone does not guarantee object-level decomposition; arguably no unsupervised formulation can provide such a guarantee as “objectness” is a context-dependent property (e.g. should a car be counted as one object or are the wheels, chassis, etc. individual objects?). Importantly, though, despite these challenges the community has managed to develop models with appropriate inductive biases that discover meaningful structure in increasingly complex datasets and that learn interpretable decompositions. Moreover, GENESIS uses the same mixture formulation as MONet whose authors conduct an analysis that shows that MONet successfully distinguishes between several objects of the same colour. To further support this, we added an additional figure to the appendix (Figure 10) to provide an example from the GQN dataset where both GENESIS and MONet successfully distinguish between two objects of the same shape and colour.
>
> Reviewer:
> “1. It would be good to show the qualitative result of the simplified Genesis-s and its qualitative result in the FID experiment.”
> Answer:
> We added Figures 8, 9, and 11 to the appendix to show qualitative results for scene generation and decomposition with GENESIS-S after training on GQN. We also added the FID scores on Multi-dSprites and GQN for GENESIS-S to Table 2. The figures illustrate that GENESIS-S struggles to capture the semantic structure of the scenes in contrast to GENESIS (and MONet).
>
> Reviewer:
> “2. Is the VAE baselines (BD-VAE and DC-VAE) trained by maximizing the ELBO? If it is the case, will the result of the FID experiment section be different if we train the VAE baseline with GECO?”
> Answer:
> The VAE baselines are also trained with GECO to enable a fair comparison.
>
> Reviewer:
> “3. Some detail of the implementation is missing, e.g. how do we model the posterior q_{\phi}(z_{k}^{c} | x, z_{1: k}^{m}).”
> Answer:
> Similar to MONet, the posterior of z_{k}^c is encoded and decoded by a `ComponentVAE’ using the same architecture as MONet but replacing ReLUs with ELUs. This is described in appendix B.1 together with the other implementation details.

---

### Official Review · AnonReviewer2 · 2019-10-24
**Official Blind Review #2**

**Rating:** 6

**Review:**

The paper proposes a generative model for images. There's a probability mask per-pixel per-component (which yields mixing probabilities), and then a set of latents per-component that yield an image. The system is tested on a set of scenes like the GQN dataset, stacks of blocks, and the multi-dsprites dataset. The system is better than MONet, although there are a few lingering questions.

Summary of positives:
+ The factoring of the image into various components is eminently sensible and more work should build in the notion of objects
+ The method is well explained, and I found it easy to understand the entire process.
+ The method does appear to perform better than MONet, and qualitatively produces good results.

Summary of negatives:
- There are a few overclaims that should be fixed: namely that the system is a "generative model of 3D visual scenes" when in reality it is a generative model of images that has a latent space that is perhaps well-positioned to match images generated by 3D scenes composed of objects.
- The method section sets up a few questions that are never answered in the paper: GENESIS--S gets introduced as a baseline to test a hypothesis, but this never gets quantitatively evaluated (indeed GENESIS-S beats GENESIS in 2/3 of the categories where they are compared head-to-head); similarly, the fact that GENESIS produces probabilities is repeatedly sold as an advantage over other methods, but it's never used.
- The experiments are weak: the aforementioned questions aren't tackled, as well as a few other issues.


Overall, I lean ever so slightly towards rejection. The results are good looking, and the method seems well-explained. However, in my view, the manuscript seems to fall short of the mark. I am not, however, strongly opposed to the paper's acceptance. If it is accepted, I would urge authors to address the issues to make their paper have maximum impact.

Big picture things:
-I'm baffled by the repeated claim that this is a generative model of 3D scenes. In the introduction "first object-centric generative model of 3D visual scenes capable of both decomposing and generating scenes". I'm worried I'm just missing something profoundly obvious and critical -- since it seems obvious to me that it's not 3D.

As far as I can tell, x here is a HxWxC image (where C is the number of channels and probably 3); the mixing probabilities are over HxW images, so it's a 2D segmentation system. It may be applied to photos rendered with a perspective camera -- but if the requirement for the system is being applied to images that come via projection, then is faster RCNN a 3D object detector? Are normalized cuts, SLIC superpixels, or any of the other myriad unsupervised segmentation methods from vision then 3D scene segmentation approaches?

-Similarly, it's not really a model for objects -- as can be seen in the multi-colored wall example in Fig 3, and Fig 8, 9. its notion of object is a region with uniform color (presumably due to the Gaussian p(x|z) ). So while it's great that it can model regions of uniform color, the system in large part seems to work (in its current form, reading the current manuscript) because the objects are all one color, and doesn't necessarily decompose the scene into objects.

Unless I'm totally missing something, this sort of claim that should be corrected very quickly. I realize that my personal bias is towards more engineering and harder data and I appreciate that there should be different operating points along the spectrum of how much information is provided. However, I think that it's worth making claims and assumptions clear, rather than claiming to solve the general problem on a specific case because of peculiarities about the specific case.


Method:
-The writing of the method is quite straightforward and written well. The system is clear to me. I largely have no complaints about the method. I do, however, have concerns about the experiments that are done to validate claims in the method section.
-GENESIS-S appears as this baseline "To investigate whether separate latents for masks and component appearances are necessary for decomposition", but then basically disappears. There are, as far as I can see, no qualitative results from it, and it basically appears once quantitatively, in S 4.3 / Table 1, where it appears to be as good if not better than GENESIS. In the appendix it's claimed that GENESIS trains faster than GENESIS-S, but if this is the selling point of GENESIS over GENESIS-S, it would be nice to have at least a little quantification of this.
-There is considerable fuss made over the fact that the system produces probabilities and this is repeatedly mentioned as an advantage over existing systems, but it is never demonstrated that these probabilities are good or useful. This leaves the reader hanging a bit.

Experiments:

Overall the experiments are a little on the weak side.

+The qualitative results are good, but the primary selling point it seems is that the system is better than MONET. This does seem clear, and GENESIS does appear quite a bit better.

- While the results on factoring scenes do appear to be good, there's no quantitative evaluation of this (although the abstract promises it). Section 4.2 essentially says that ARI is bad, without demonstrating in a compelling way that it is. The paper says that this behavior can be seen in Appendix D, but this only shows the oversegmentation and not that this dramatically throws off performance -- I'd expect a figure showing a reversal of expectations -- a clearly worse result getting a clearly better ARI metric.

Additionally, computer vision has been evaluating unsupervised segmentation for close to two decades (BSDS came out in 2001). The authors should look at the metrics in e.g., "Contour Detection andHierarchical Image Segmentation" Arbelaez et al. PAMI 2010. I realize that coming up with metrics is hard, but I think the burden is on the authors to find the metrics to show their conclusions quantitatively.

- The tower stability/height/view experiments seem incomplete. I find them quite interesting, but I'm not entirely sure of what to draw from it. The paper obliquely comments that the Groth paper gets better results with a more complex backbone network and blames it on lack of augmentation, use of a subset of the data, and a reduced resolution. But why not just do straightforward like train a MLP on an even further spatially downsampled image? Random here isn't a particularly compelling baseline or reference point.

Small stuff that doesn't affect my review:
1) It might be worth explicitly pointing out that p_\theta(x_k,z_k^c) is just a Gaussian soon after Eqn 3. It's written after equation 5, but without making the p_\theta explicit. This may help some readers.
2) The tower stability is done with a one-hidden-layer (512) MLP. I'm of the personal opinion that readouts on latent variables are more informative if it's a linear transformation (since this can't do much heavy lifting).
3) The abstract promises semi-supervised learning -- is this the tower experiment? This strikes me more as transfer learning.

-----------------------------

Post review update: I have read the authors' responses, and found them thoughtful and to have answered my questions. I'm happy to accept the paper and would encourage the AC to do so.

**Experience Assessment:**

I have published in this field for several years.

**Review Assessment: Checking Correctness Of Derivations And Theory:**

I assessed the sensibility of the derivations and theory.

**Review Assessment: Checking Correctness Of Experiments:**

I carefully checked the experiments.

**Review Assessment: Thoroughness In Paper Reading:**

I read the paper thoroughly.

---

> ### Author Response · Authors · 2019-11-09
> **Reply to reviewer 2 from the authors - part 1/2**
>
> We thank the reviewer for the thorough feedback and constructive criticism. We updated the paper to address several shortcomings: we rephrased our main claim, we added further qualitative and quantitative results for GENESIS-S, we computed ARI scores for GENESIS and MONet for five different random seeds, and we added ARI scores to Figures 12 and 13 (numbering in updated manuscript) to illustrate the limitations of the ARI metric. Further comments follow below.
>
> Reviewer:
> “I'm baffled by the repeated claim that this is a generative model of 3D scenes.”
> Answer:
> There appears to be a misunderstanding regarding what is meant by a “generative model of 3D visual scenes”. We intend to refer to 2D rendered images of environments with 3D spatial structure (i.e. images from the GQN dataset and ShapeStacks). The majority of previous works on structured latent-variable models (AIR, NEM, R-NEM; IODINE is a notable exception) only use datasets such as Multi-MNIST, Multi-dSprites, Tetrominoes, or ATARI games which lack 3D spatial structure, non-uniform backgrounds, etc. This is elaborated on in the related work section, but we will rephrase the affected parts to “rendered 3D scenes” to avoid ambiguity.
>
> Reviewer:
> “Similarly, it's not really a model for objects -- as can be seen in the multi-colored wall example in Fig 3, and Fig 8, 9. its notion of object is a region with uniform color (presumably due to the Gaussian p(x|z) ). So while it's great that it can model regions of uniform color, the system in large part seems to work (in its current form, reading the current manuscript) because the objects are all one color, and doesn't necessarily decompose the scene into objects.”
> Answer:
> This is an interesting point of discussion. We would like to note that there is no unique assignment of  “objectness”, e.g. referring to Figure 3: should a wall be considered as one object, or should tiles in the wall be considered as different objects? Unsupervised structured latent-variable models do not contain any explicit constraints with regards to what constitutes an “object”. How scenes are decomposed depends merely on the structure of data and the inductive biases in the model. Interestingly, though, despite these challenges the community has managed to develop models that manage to decompose scenes into semantically meaningful components without explicit supervision.
> Overall, we agree that it might be more accurate to talk about “scene components” or “entities”; it is also unusual to refer to things like the floor or the sky as objects. However, we decided to adopt the same terminology as used by the community in previous works, such as MONet and IODINE. We were hoping our footnote in the introduction (“We use the terms ‘object’ and ‘scene component’ synonymously in this work.”) clarifies this. Nevertheless, we are open to rephrasing these parts if the reviewer is not satisfied with this justification.
> Finally, we agree that current methods rely heavily on colour and texture as cues for learning to decompose scenes and that scaling these methods to more complex datasets such as real-world images remains an open challenge. Nevertheless, we believe that by providing a fully probabilistic model of scenes that captures relationships between components and a more scalable, probabilistic attention mechanism, GENESIS overcomes key limitations of previous works (independent priors for scene components; inefficient, deterministic attention or expensive iterative inference) and therefore makes valuable contributions towards developing more general methods.
>
> Reviewer:
> “GENESIS-S appears as this baseline "To investigate whether separate latents for masks and component appearances are necessary for decomposition", but then basically disappears.”
> Answer:
> We added Figures 8 and 9 which show qualitative results for scene generation with GENESIS-S after training on GQN; we added the FID scores for GENESIS-S on Multi-dSprites and GQN to Table 2; and we added Figure 11 which show the step-by-step decomposition of three images from the GQN dataset.
>
> Reviewer:
> “In the appendix it's claimed that GENESIS trains faster than GENESIS-S, but if this is the selling point of GENESIS over GENESIS-S, it would be nice to have at least a little quantification of this.”
> Answer:
> While it is true that GENESIS trains faster than GENESIS-S, the main selling point of GENESIS over GENESIS-S is that it does a better job at decomposing scenes into meaningful objects and generating interpretable components (as illustrated in the additional Figures mentioned above). We therefore removed the “faster training” claim as it misses the point and clarified the benefit of GENESIS over GENESIS-S in the main body of the paper (rather than the appendix).

---

> > ### Author Response · Authors · 2019-11-09
> > **Reply to reviewer 2 from the authors - part 2/2**
> >
> > Reviewer:
> > “There is considerable fuss made over the fact that the system produces probabilities and this is repeatedly mentioned as an advantage over existing systems, but it is never demonstrated that these probabilities are good or useful.”
> > Answer:
> > The probabilistic formulation enables GENESIS to generate new scenes in a principled fashion by sampling from the prior. That said, we found this comment to be thought provoking and we will investigate how to further illustrate the benefits of the probabilistic formulation.
> >
> > Reviewer:
> > “While the results on factoring scenes do appear to be good, there's no quantitative evaluation of this (although the abstract promises it). Section 4.2 essentially says that ARI is bad, without demonstrating in a compelling way that it is. The paper says that this behavior can be seen in Appendix D, but this only shows the oversegmentation and not that this dramatically throws off performance -- I'd expect a figure showing a reversal of expectations -- a clearly worse result getting a clearly better ARI metric.”
> > Answer:
> > The abstract does not promise a quantitative evaluation of scene decomposition. Yet, we updated section 4.2 and added means and standard deviations for the ARI scores after training Genesis and MONet with five different random seeds on ShapeStacks. Despite its limitations, the ARI metrics still gives some indication of segmentation quality. We also added ARI scores to the captions of Figures 12 and 13, whereby Figure 13 shows a qualitatively worse segmentation achieving a better ARI score to illustrate the aforementioned limitations.
> >
> > Reviewer:
> > “The authors should look at the metrics in e.g., "Contour Detection andHierarchical Image Segmentation" Arbelaez et al. PAMI 2010.”
> > Answer:
> > We thank the reviewer for pointing us to this work and we will investigate this.
> >
> > Reviewer:
> > “The tower stability/height/view experiments seem incomplete. I find them quite interesting, but I'm not entirely sure of what to draw from it.”
> > Answer:
> > We do not regard this section as a central component of the manuscript, but rather a set of additional, exploratory experiments to verify that object-centric representations are indeed beneficial for downstream tasks. The experiments give some support to this hypothesis, though a more comprehensive study would be required to make strong claims.
> >
> > Reviewer:
> > “The abstract promises semi-supervised learning -- is this the tower experiment? This strikes me more as transfer learning.”
> > Answer:
> > Yes, it is. This approach corresponds to the “Latent-feature discriminative model (M1)” from “Semi-supervised Learning with Deep Generative Models” by Kingma, Mohamed, Rezende, and Welling (NeurIPS 2014).

---

> > > ### Author Response · Authors · 2019-11-15
> > > **Update from the authors**
> > >
> > > We followed the reviewer’s suggestion and adopted the segmentation covering (SC) metric from "Contour Detection and Hierarchical Image Segmentation" (Arbelaez et al., PAMI 2010) for quantifying scene decomposition performance.
> > >
> > > The paper has been updated in three locations:
> > > - Quantitative results using the SC metric were added to section 4.2. We also re-computed the ARI scores on the same 300 images from the ShapeStacks test set as used for computing the SC scores for direct comparability, noting that the authors of IODINE also use 300 images for computing ARI scores.
> > > - A short section was added to the appendix to explain how the SC metric is computed.
> > > - The captions of Figures 12 and 13 were updated to also include the SC metric for the shown examples, illustrating how the ARI achieves a better score on a qualitatively worse segmentation while the SC metric reflects the segmentation quality appropriately.

---

### Decision · Program_Chairs · 2019-12-19

**Decision:**

Accept (Poster)

**Comment:**

This paper offers a new method for scene generation.  While there is some debate on the semantics of ‘generative’ and ‘3d’, on balance the reviewers were positive and more so after rebuttal.  I concur with their view that this paper deserves to be accepted.